# Characteristics Related to Choice of Obstetrician-Gynecologist among Women of Ethiopian Descent in Israel

**DOI:** 10.3390/healthcare8040444

**Published:** 2020-10-30

**Authors:** Avi Zigdon, Gideon Koren, Liat Korn

**Affiliations:** 1Department of Health Systems Management, School of Health Sciences, Ariel University, 40700 Ariel, Israel; aviz@ariel.ac.il; 2Adelson Faculty of Medicine, Ariel University, 40700 Ariel, Israel; gidiup_2000@yahoo.com; 3Motherisk Israel Program, Shamir Medical Center, 70300 Be’er Ya’akov, Israel

**Keywords:** obstetrician-gynecologist, gender, Jewish Ethiopian immigrant, patient satisfaction

## Abstract

Background: Patient satisfaction with the quality of health care services is complex with many known factors impacting upon satisfaction, among them the choice of physician. Previous studies examined characteristics of a woman’s choice of gynecologist, but information regarding reasons for these choices among women of Ethiopian descent is lacking. The objective of this study was to identify characteristics related preference of an obstetrician-gynecologist based on gender among women of Ethiopian descent. Method: Analysis of anonymous self-reported questionnaire distributed to 500 women of Ethiopian descent who visited an obstetrician-gynecologist at least once in the past three years (Mean age 29.5; SD = 8.2). Trust in physician was examined using the TPS scale; service quality was checked using the SERVQUAL; and the 5Qs model was used to measure patient’s satisfaction of health care. Results: Very religious (84.1%) and religious (53.6%) women of Ethiopian descent were more likely to prefer a female obstetrician-gynecologist compared to traditional (39.3%) or secular (34.4%) women (*p* < 0.001). Participants had higher probability to prefer a male gynecologist if they had more employment hours (OR = 3.57, 95% CI = 1.72–7.42, *p* < 0.001), and the responsiveness of the physician was less important to them (OR = 0.77, 95% CI = 0.60–0.99, *p* < 0.05). Age of participants, country of birth, years in Israel, family status, being a mother, education or health status were not associated with choosing a female obstetrician-gynecologist. Religious women would prefer to wait longer for a female gynecologist due to modesty imperatives, even at the cost of compromising their health as a result of waiting longer. Conclusions: The findings of this research highlight the importance of accessibility to female obstetrician-gynecologists for women of Ethiopian descent and demonstrate that determinants in the host population rather than immigrant’s past culture, affect the women’s decision. This study demonstrates the importance of the health care system in Israel to enable more female obstetrician-gynecologists to treat women of Ethiopian descent.

## 1. Background

Choosing a physician is a complex process in public health systems, affected by visible and hidden activities of the health funds and regulatory processes [1]. The ability to choose a physician was found to greatly increase patient satisfaction and the likelihood of continuity of care [2,3,4]. Choice of physician has been associated with a variety of determinants [3] attributed to four main categories: communication and physician-patient relations [4,5], physician professionalism, personal aspects of the patient and physician [6] and development of web-based information for physician ratings [7,8]. Anderson and Dedrick [9] demonstrated that satisfaction is a matter of trust in the physician. Kitapci and colleagues suggested five components for quality of health services in the SERVQUAL quality model [10]: reliability, assurance, responsiveness, empathy and tangibles. Another model is the 5Q construct, also suggesting five different components [11]: quality of object, processes, infrastructure, interaction and atmosphere.

Selecting an obstetrician-gynecologist is mainly influenced by physician’s gender and professionalism [12,13,14,15,16,17], and by religious beliefs of the patient [12]. Studies have documented how religious beliefs influence women’s health choices such as abortion [18] and use of contraceptives [19]. Research indicates that physician’s gender does affect choice of gynecological treatment [20,21], but characteristics such as communication and personal style of the physician take precedence in selecting a gynecologist [21]. While refugee women exhibit similar gynecologic needs to the broader population [22], their behavior tends to be influenced by different factors, and their use of health services is affected by their culture of origin [23].

In the 1970s, Ethiopian Jews began to immigrate to Israel and as of the end of 2017, the population of Ethiopian immigrants in Israel numbered 147,000, of whom 87 thousand were Ethiopian-born and 61.7 thousand were born in Israel. Approximately 50% of them were women [24]. The sharp transition from an environment of African traditional to a Western health perception placed Ethiopian women in a high-risk group when considering health [25]. However, health behavior of Ethiopian women has similar characteristics to that described in the literature for immigrant and ethnic minority populations and they may have lower standards of care [26], suffering from more discrimination and unfairness in health services [27] and leading to lower overall health. The utilization of primary health care by sub-Saharan African immigrants in Norway differs from that of the host population [28], preferring informal health services, with focus on religious healing instead of Western practices such as those found in older Somali immigrants in Finland [29].

As of 2019, only 20% of Israeli obstetrician-gynecologists were female [30], making it difficult for women to be cared by a female. To our knowledge, no studies have been published on the choice of obstetrician-gynecologists among Jewish Ethiopian women who immigrated to Israel, highlighting the importance of the present study.

Little is known about health preferences of disadvantaged populations [13,14,22] such as minorities and refugees. By contrast, quite a lot is known about choices of dominant populations in different countries [16,17,18]. Naturally, disadvantaged populations had not been appropriately investigated and therefore are likely to receive services that are less suitable for them, thus creating a situation where they more commonly have unmet needs [13,14,16,17,22].

Thus, our objective was to identify these characteristics and their associations among women of Ethiopian descent living in Israel as a model of a disadvantaged population. Our hypotheses were that religious women will tend to prefer female obstetrician-gynecologists and that different aspects of physician care satisfaction will be associated with the physician’s gender choice.

## 2. Methods

### 2.1. Study Design, Setting and Population

Data were collected by anonymous self-reported printed questionnaires delivered face to face in Hebrew or Amharic, between June 2017 and June 2018. A convenience sampling was used with a snowball method, the survey team directly contacting participants based on prior familiarity with the team or with participants through a second and third referral to reach only Ethiopian women immigrants or women of Ethiopian descent. A sample considered sufficiently large for statistic strength, and the maximum was reachable in time and budget constraints. The study was approved by the Ethics Committee of Ariel University(AU-AZ-20180612).

The sample included 500 Ethiopian women immigrants or women of Ethiopian descent from all parts of Israel, aged 18 or older, who visited an obstetrician-gynecologist at least once in the past three years (Mean age 29.5; SD = 8.2, Min-Max 18–75). This sample size is considered sufficiently large for statistic strength, and the maximum that could be reachable within time constraints. Participants were advised that they were free to choose not to participate in the study and to cease their participation at any time. Table 1 presents the sample’s characteristics. Most of the participants were born in Israel (62.4%), and most of those born in Ethiopia had lived in villages (55.3%) and immigrated to Israel during their childhood (75.8% before 12 years of age). More than half of the women sampled were single (60.8%), considered their religiosity as traditional (51.3%), had attained higher education (68.2%), reported very good health (73.7%) and reported being at least 80% employed (72.8%).

### 2.2. Formal Instruments

The structured questionnaire contained 4 sections: (1) “Trust in Physician Scale” (TPS) [31], with 11 items on a scale designed to measure interpersonal patient trust in physicians relating to primary care services; (2) The SERVQUAL quality models [32], which contain five dimensions of service quality; (3) 5Qs model, a comprehensive model of patient satisfaction with healthcare providers [33]; and (4) socio-demographic questions.

### 2.3. Description of Measures

Country of birth—Israel, Ethiopia; Type of residence in Ethiopia—City, village or else. Birth year—participants were asked to report their year of birth. Age was calculated from their year of birth and scale was dichotomized by the median (34) for further analysis. Year of immigration to Israel—participants were asked to report their year of immigration; Religion—Participants were asked how they defined their religiosity—values were 1. Secular, 2. Traditional (Commonly means that they believe in a deity but do not strict in keeping the commandments of religion), 3. Religious, 4. Very religious. Family status—values were 1. Single, 2. Married 3. Divorced, 4. Widower, 5. In relationship. For further analysis, these values were divided into 1. Single, 2. Married or in relationship, 3. Divorced or widower; Education—participants were asked which academic level they reached in their studies: 1. Elementary or middle school, 2. High school/religious yeshiva, 3. Higher education, 4. University, 5. Other educational setting, 6. I didn’t finish elementary/did not study at all. For further analysis values were dichotomized: 1. High school or less (values 1,2,6) and 2. Higher education (values 3,4,5). Health status—participants were asked how they defined their health status. Values were: 1. Bad, 2. Not good, 3. Not so good, 4. Good, 5. Very good. For further analysis values were dichotomized by the median (1): 1. Very good, and 2. Less than very good; Years in Israel—was calculated from birthplace, age and age of immigration. Scale (16–69) was dichotomized by median (27 years) for further analysis.

Trust in Physician Scale—TPS [31]. This scale contains 11 items measuring interpersonal trust of patients on gynecologist physician. For example: “My doctor is usually considerate of my needs and puts them first” or “I trust my doctor to tell me if a mistake was made about my treatment”. Values ranged on five points Likert scale from 1—not important at all to 5—very important. Cronbach alpha (α = 0.85). Items were summed and ranged from minimum of 18—low trust till maximum of 55—high trust, then dichotomized by the median (45) to reflect high and low group levels of trust.

Quality of services scale—SERVQUAL quality model [32]. This scale (32–65) contains 15 items (Cronbach alpha α = 0.78) reflecting the amount of satisfaction from service quality. Values ranged on five points Likert scale from 1—not important at all to 5—very important. The scale contains five dimensions: 1. Reliability—one item regarding the importance of my trust toward my gynecologist. 2. Assurance—the ability of my gynecologist to inspire trust and confidence. Four items (“The level of knowledge of my gynecologist”/“The treatment offered is appropriate”/“The variety of treatments offered (such as pregnancy monitoring and routine pregnancy tests, system review, back translucency, amniotic fluid, etc.)”/“The degree of security you feel when treating”. (Cronbach alpha α = 0.61; 10–20). 3. Responsiveness—the willingness to help customer and provide prompt service. Three items regarding time responsiveness, availability and understanding my needs (Cronbach alpha α = 0.61; 6–15). 4. Empathy—caring, individualized attention. Two items reflected empathy regarding personal attitude and courtesy (Cronbach alpha α = 0.65; 3–10). 5. Tangibles—physical facilities, appearance of personnel and the equipment. Five items reflected characteristics of the gynecologist such as gender, age experience and good recommendations (Cronbach alpha α = 0.61; 3–10).

Satisfaction from quality of health services scale—This scale, based on 5Qs model for patient satisfaction from health care providers [33], contains 17 items (Cronbach alpha α = 0.85; 44–85). Values ranged through a five-point Likert scale from 1—not important at all—to 5—very important. The scale contains 5 quality dimensions: 1. Quality of object–relates to the clinical procedures and the purpose of the visit. Three items: “While staying in the clinic, there is a sense of security from the services offered”/“The degree of well-being you feel in the women’s clinic”/“The clinic’s ability to treat you in the way you expect” (Cronbach alpha α = 0.78; 6–15). 2. Quality of processes—functional quality of health care organization. Two items reflected that regarding the amount of waiting time when scheduling the appointment and amount of waiting time in line for examination/treatment (Cronbach alpha α = 0.89; 2–10). 3. Quality of infrastructure—measures the essential and basic resources needed to perform the health care services. Three items regard treatment room temperature, clinic staff outside appearance and cleanliness of treatment rooms (Cronbach alpha α = 0.74; 3–15). 4. Quality of interaction–communication and understand the patient’s reflected by three items: accessibility, speed and ease of entry into the clinic, Linguistic communication ability with the gynecologist, and doctor’s skills providing the service. (Cronbach alpha α = 0.64; 6–15).

Needs; 5. Quality of atmosphere– atmosphere in a specific environment where they cooperate and operate reflected by four items regarding explanations given to you about the treatments recommended for you, the ease with which the clinic can be obtained by phone, the responsiveness of the staff attending to your needs and the information you receive about your condition (Cronbach alpha α = 0.78; 9–20). Fifteen items were grouped after factor analysis, leaving 2 items, “courtesy of clinic staff” and “services provided at the clinic are performed as required”, out of the five dimensions but contributing to reliability of the scale.

### 2.4. Dependent Variables

Gynecologist female/male preferred—participants were asked two questions on the importance of their gynecologist’s gender. Values ranged on a five point Likert scale from 1—not important at all—to 5—very important. Gynecologist gender—participants were asked the sex of their gynecologist: female or male.

### 2.5. Statistical Analysis

The quantitative data were analyzed using IBM SPSS-21 (IBM, Armonk, NY, USA). Pearson chi-square for independency was performed to examine the relation between religion to three variables of gender choice. Table 1 shows descriptive statistics by frequencies presented in percentage of each variable’s categories. Table 2 presents a cross-tabulation frequency between religion and preference of gender gynecologist with Pearson chi-square testing for differences between groups. Table 3 presents outcomes from hierarchic logistic regression to predict preference for a male obstetrician-gynecologist in two phases. Results express OR (odds ratio) and 95% CI (confidence interval) and significance. Predicting female obstetrician-gynecologist preference or predicting gynecologist gender were checked during data analysis, and findings supported the same trend but with lower predicting power, so male obstetrician–gynecologist was our choice for presentation.

## 3. Results

Pearson chi-square for independency shows relation between religion to three variables of gender choice representing significant relation between these variables (*p* < 0.001). Religious (53.6%) and very religious (84.1%) women were more likely to prefer a female obstetrician-gynecologist compared to secular (34.4%) or traditional (39.3%) women. In addition, the likelihood of their obstetrician-gynecologist to be female was higher (76.5% and 90.9% compared to 43.8% and 44.9%, respectively). As expected, the findings for male obstetrician-gynecologists showed the opposite direction with lower percentage of very religious (2.3%) and religious women (17.4%) and higher frequencies of secular (32.6%) and traditional women (33.6%) who preferred a male obstetrician-gynecologist.

Table 3 presents outcomes from hierarchic logistic regression to predict male obstetrician-gynecologist preference in two phases: first, for socio-demographic characteristics, and second, for significant socio-demographic characteristics from phase 1 and the independent variables of the study. Explained variance rose from 13.4% in phase 1 to 35.7% in phase 2. In phase 1, out of nine variables only religion (OR = 0.25, 95% CI = 0.13–0.56, *p* < 0.001) and employment (OR = 2.87, 95% CI = 1.50–5.48, *p* < 0.001) were significant sociodemographic characteristics that predicted male obstetrician-gynecologist preference, which also stayed significant in phase 2 (OR = 0.22, 95% CI = 0.09–0.56, *p* < 0.01 for religion and OR = 3.57, 95% CI = 1.72–7.42, *p* < 0.001) for employment). Of the independent variables in phase 2, only responsiveness (OR = 0.77, 95% CI = 0.60–0.99, *p* < 0.05) and tangibles (OR = 1.51, 95% CI = 1.32–1.73, *p* < 0.001) were significant in predicting male obstetrician-gynecologist preference. Trust in physician, assurance, empathy and different dimensions of quality of services were not significant in predicting male obstetrician-gynecologist preference. Findings in this table show that Israeli women of Ethiopian descent will have higher probabilities of preferring a male obstetrician-gynecologist if they are secular rather than religious, are employed more than 80%, the responsiveness of the physician is less important to them, and if they rated tangibles as more important.

## 4. Discussion

Although forty years have elapsed since the mass immigration of Ethiopian Jews to Israel, Jewish Ethiopian women are defined as a high-risk group as related to wellness [25]. Physician gender, professionalism [9,10,11,12,13,14] and religious beliefs [9] were found to be factors influencing the choices of migrant women. To our knowledge, no studies have been published to date on choice of obstetrician-gynecologist among Jewish Ethiopian women who immigrated to Israel or women of Ethiopian descent. The present study sought to identify the characteristics related to this choice using validated research tools [9,10,11]. Although the sampling method was based on snowball-type recruitment, a matching proportion was found between sample age, religion, marital status, education and average number of children per family for the general Jewish Ethiopian population in Israel [24].

One of the strongest associations found in our study is between obstetrician-gynecologist gender choice and religion. The more religious the Jewish women of Ethiopian descent, the more they tended to choose a female obstetrician-gynecologist. As suggested [9], choosing an obstetrician-gynecologist is influenced by religious beliefs, and these might constitute a barrier against a female exposing her genitals to a male gynecologist or obstetrician. The relationship between religion and gynecologist gender choice becomes clearer if looked through the prism of female religious modesty [22]. This is all the more so for religious modesty codes that prohibit women from exposing their arms and/or hair in the fear that they may inadvertently excite men. Thus, the modesty-based choice of obstetrician-gynecologist is far more meaningful for religious or ultra-orthodox women compared to their secular counterparts. For religious women, the potential for provocation caused by lack of modesty infuses a variety of life aspects and decisions: from buying outfits to movements and sitting positions to healthcare choices. However, secular women are not socialized to habituate themselves in this totalizing manner. Commenting on the social construction of the obstetrician-gynecologist’s clinic, Emerson noted [31] the clinic can help patients gain confidence or increase embarrassment. In the latter case, Emerson referred to an example in which a woman delayed her first visit to her gynecologist until her eighth month of pregnancy.

The question of obstetrician-gynecologists gender provoked feelings of embarrassment [32] that can be attributed to cultural, traditional, and religious undercurrents in Egyptian society. These constitute a barrier against a female exposing her genitals to a male gynecologist or obstetrician. Considering the rise in migration across the globe, many studies have investigated differences between immigrant and non-immigrant women’s health, with some finding gaps in health between the two populations [33,34]. Other studies reported that health disparities among migrant women are affected by physician characteristics [35], education [36], information access, preference for physicians with an immigrant background, financial barriers and long wait times [37]. As noted in the reviewed literature, the sharp transition from non-Western to Western style healthcare placed Ethiopian women in a high-risk group for women’s health [25,26,27,28,29].

Another significant finding was the association of employment and obstetrician-gynecologist gender preference. We found that the more hours a woman was employed, the more likely her choice of a male obstetrician-gynecologist. This finding has not been previously described in the published literature. We found that women of Ethiopian descent will prefer a male obstetrician-gynecologist if she values his professionalism, experience and positive recommendations, matching up with previous research [9,10,11,12,13,14]. We also found that it was less important to Jewish Ethiopian women that male obstetrician-gynecologist understand their needs. We assume that this choice may be due to the low availability of female gynecologists in Israel, with a rate of only 20% [24]. Another finding is that wait times for both male and female gynecologists mattered for all groups of women of Ethiopian descent. However, religious women would prefer to wait longer for a female gynecologist due to modesty imperatives [22], even depriving their health as a result of waiting longer. As suggested before [38] immigration involves behavioral shift and adaptation to the dominant culture to reduce stress and conflicts which are expected to occur while transforming to other culture. The Israeli health system should make this process easier for women of Ethiopian descent population in order to reduce the embarrassment that may occur in this encounter, have more convenient solutions and facilitate this process that is intensifying with cultural difficulty.

Several limitations of this study need to be acknowledged. Snowball sampling method makes it difficult to represent all Israeli women of Ethiopian descent. Comparing sociodemographic characteristics shows great similarities between the sample and the Ethiopian population in Israel, few parameters, such as age, religion, marital status, education, and average number of children per family, were found to match those in the general population [24]. The large proportion of single women in this survey and non-mother group, although matching the Ethiopian population in Israel, could affect the outcomes. Choosing a gynecologist is a major issue if you are pregnant and is different if you have no children. Only a quarter of this sample had children, and fertility might be a major issue in choosing an obstetrician-gynecologist, but not necessarily his/her gender. One other limitation of this study is that based on the complexity of tangibility measure as defined by Kitapci et al. [10], both doctor and clinic characteristics should be included, whereas our study included only the former.

## 5. Conclusions

In the migration of Jewish Ethiopian women to Israel, cultural assimilation is yet to be seen in health service consumption habits. When choosing an obstetrician-gynecologist, they take into consideration the gender of the gynecologist. The academic literature supports these findings, with religious women tending to prefer female obstetrician-gynecologist. The more religious the woman, the greater her preference for a female gynecologist [9] due to modesty concerns [22]. Nevertheless, the findings of this study show that when women of Ethiopian descent are assimilated in the host population and join the local workforce, choice of gynecologist gender is altered by other parameters related to Western lifestyle habits (e.g., increased work hours). These findings might suggest that the faster the immigrant acculturation, the higher the number of parameters of consuming health services affected by the host population rather than immigrant’s past culture. If shyness and modesty cause women not to be seen by a physician, this study demonstrates the need of the health care system to offer more female obstetrician-gynecologists to the population of women of Ethiopian descent in Israel. As there are large volumes of immigration from Africa to Europe and North America, similar trends and needs should be explored.

## Figures and Tables

**Table 1 healthcare-08-00444-t001:** Sample description.

Variables	Values	*n*	Valid Percent
Total		500	100.0
Age	27 or younger 28 or older	275 223	55.2 44.8
Country of birth	Israel Ethiopia	299 180	62.4 37.6
Religion	Secular Traditional Religious Very religious	130 256 69 44	26.1 51.3 13.8 8.8
Family status	Single Married/in relationship Divorced Widower	299 171 19 3	60.8 34.7 3.9 0.6
Number of children	0 1 2 3 4 or more	373 21 28 38 40	74.6 4.2 5.6 7.6 8.0
Education	High school or less Higher education	157 337	31.8 68.2
Health status	Not good Not so good Good Very good	1 8 122 368	0.2 1.6 24.4 73.7
Age of immigration to Israel	0–5 6–12 13–17 18 and above	76 74 32 16	38.4 37.4 16.2 8.1
Percent of employment	80% or less 81–100%	101 271	27.2 72.8

**Table 2 healthcare-08-00444-t002:** Frequency distribution of obstetrician-gynecologist gender preference by religiosity among Israeli Ethiopian women.

Variables		Obstetrician-Gynecologist Female Preferred		Obstetrician-Gynecologist Male Preferred		My Obstetrician-Gynecologist Gender Is		*n*
		Important	Not Important	Important	Not Important	Female	Male	
Religiosity	Secular	34.4%	65.6%	32.6%	67.4%	43.8%	56.2%	130
	Traditional	39.3%	60.7%	33.6%	66.4%	44.9%	55.1%	254
	Religious	53.6%	46.4%	17.4%	82.6%	76.5%	23.5%	68
	Very religious	84.1%	15.9%	2.3%	97.7%	90.9%	9.1%	44
	Total	44.0%	56.0%	28.3%	71.7%	53.0%	47.0%	496

Pearson chi-square for independence between religion to three variables of gender preferences *p* < 0.001.

**Table 3 healthcare-08-00444-t003:** Predicting preference for a male obstetrician-gynecologist from study variables: outcomes from hierarchic logistic regression. (Reference group = not important that my obstetrician-gynecologist be male).

Variables		Values	Phase 1		
			OR	95% CI	
				Lower	Upper
Phase 1 Socio-Demographic	Age	1 = 27 or younger, 2 = 28 or older	2.30	0.69	7.58
	Country of birth	1 = Israel, 2 = Ethiopia	1.22	0.66	2.24
	Religion	1 = Secular, 2 = Traditional, 3 = Religious, 4 = Very religious	**0.25 *****	**0.13**	**0.56**
	Family status	1 = single, divorced, widower, 2 = married, in relationship	0.93	0.50	1.71
	Children	1 = no, 2 = yes	1.21	0.55	2.64
	Education	1 = High school or less, 2 = Higher education	0.94	0.56	1.59
	Health status	1 = less than very good, 2 = very good	1.32	0.73	2.39
	Years in Israel	1 = 27 years or less, 2 = 28 years or more	0.39	0.13	1.18
	Percent of employment	1 = less than 80%, 2 = 81% or more	**2.87 *****	**1.50**	**5.48**
*n*			**335**		
Nagelkerke R Square			**13.5%**		
**Variables**		**Values**	**Phase 2**		
			**OR**	**95% CI**	
				**Lower**	**Upper**
Phase 2 Socio-demographic and independent variables	Religion	1 = Secular, 2 = Traditional, 3 = Religious, 4 = Very religious	**0.22 ****	**0.09**	**0.56**
	Percent of employment	1 = less than 80%, 2 = 81% or more	**3.57 *****	**1.72**	**7.42**
	Trust scale	1 = low trust, 2 = high trust	1.14	0.61	2.12
	Reliability	Lowest—less important…. highest—more important	0.67	0.41	1.09
	Assurance		1.04	0.82	1.33
	Responsiveness		**0.77 ***	**0.60**	**0.99**
	Empathy		0.81	>0.62	1.07
	Tangibles		**1.51 *****	**1.32**	**1.73**
	Quality of object		1.01	0.83	1.23
	Quality of process		1.06	0.82	1.37
	Quality of Infrastructure		0.96	0.83	1.10
	Quality of Interaction		1.20	0.93	1.55
	Quality of atmosphere		0.86	0.71	1.05
*n*			**310**		
Nagelkerke R Square			**35.7%**		

Significant values bolded, * *p* < 0.05, ** *p* < 0.01, *** *p* < 0.001.

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
