# Peer review of "Characteristics Related to Choice of Obstetrician-Gynecologist among Women of Ethiopian Descent in Israel"

_healthcare, 2020, doi:10.3390/healthcare8040444_

Round 1
Reviewer 1 Report
I would like to reiterate that I cannot assess the statistical analyses.
The draft article is interesting and well written. Some improvements should be made:
In the background section, references should be used in a more stringent way. E.g. the statement where ref. 29 is being used, a study that only deals with Somalis is quoted as being relevant generally for immigrants. Statements in the reference articles that are in turn references are quoted (as in ref. 28), with the danger of bias.
The strongest part is the description of the methods and the results. But "traditional" as a category of religiousness should be defined. Number and percentage of respondents with higher education is missing.
in the conclusion section, it should be taken care that the statements are actually based on what the article has described, and not bring in new and unsubstantiated ones. I do not see that the religiousness of the provider has been analysed in the article, but it appears here. The references seem to be inappropriate. The statement that "the faster the immigrant acculturation, the higher the number of parameters of consuming health services affected by the immigrant acculturation rather than immigrants' past culture" appears as far too general than what the results can substantiate.
Author Response
Response to Comments and Suggestions for Authors
Reviewer 1:#1
I would like to reiterate that I cannot assess the statistical analyses. The draft article is interesting and well written. Some improvements should be made: In the background section, references should be used in a more stringent way. E.g. the statement where ref. 29 is being used, a study that only deals with Somalis is quoted as being relevant generally for immigrants. Statements in the reference articles that are in turn references are quoted (as in ref. 28), with the danger of bias.
Authors response:
We thank the reviewer for the critical reading, for the time and effort and for the excellent comments. We have reread the entire paper, emphasizing details from reviewed papers in the introduction section, to better express the specific knowledge they contributed. We have made a few changes in the text accordingly to avoid the possibility of bias.
Reviewer 1:#2
The strongest part is the description of the methods and the results. But "traditional" as a category of religiousness should be defined. Number and percentage of respondents with higher education is missing.
Authors response:
Details regarding "traditional" category of religiousness were added as follow (Page 4, Description of measures section): Religion- Participants were asked how they defined their religiosity- values were- 1. Secular, 2. Traditional (Commonly meaning is that they believe in a deity but do not strict in keeping the commandments of religion), 3. Religious, 4. Very religious. Higher education details were added to the table 1 (page 15) as follow: Number- 337, Percentage- 68.2.
Reviewer 1:#3
in the conclusion section, it should be taken care that the statements are actually based on what the article has described, and not bring in new and unsubstantiated ones. I do not see that the religiousness of the provider has been analysed in the article, but it appears here. The references seem to be inappropriate. The statement that "the faster the immigrant acculturation, the higher the number of parameters of consuming health services affected by the immigrant acculturation rather than immigrants' past culture" appears as far too general than what the results can substantiate.
Authors response:
We have reread the conclusion section and have edited statements that were not outcomes of this paper, and refined the wording as requested.
Reviewer 2 Report
This is a very interesting and well presented paper about the choice of gynaecologist in Ethiopian women in Israel. It is important as there are only around 20% of women gynaecologists available in Israel, and religious women may delay attendance, rather than to see a male.
My only comment is the wording in the Abstract where the authors talk about 'gynaecologist choice'. This is not the gynaecologist's choice but rather the woman's choice. A little re-phrasing would help.
My concerns are with the wording of the Abstract which was not well written.I have attached the changes I would suggest.

Author Response
Comments and Suggestions for Authors
Reviewer 2:#1
This is a very interesting and well presented paper about the choice of gynaecologist in Ethiopian women in Israel. It is important as there are only around 20% of women gynaecologists available in Israel, and religious women may delay attendance, rather than to see a male. My only comment is the wording in the Abstract where the authors talk about 'gynaecologist choice'. This is not the gynaecologist's choice but rather the woman's choice. A little re-phrasing would help.
Authors response:
We thank the reviewer for the critical reading, for the time and effort. Thank you for the excellent suggestions. We altered the text as suggested.
Reviewer 2:#2
My concerns are with the wording of the Abstract which was not well written.
I have attached the changes I would suggest.
Authors response:
Corrections have been made as suggested. Thank you.
Reviewer 3 Report
This is well-structured papers that makes a sound argument. As a feminist sociologist of religion, I did not find its main argument to be particularly novel and the modesty-concerns of religious women are common knowledge.
However this paper does indeed make a valuable contributing in quantifying these concerns and making recommendations that are based on statistical information rather that qualitative data.
I recommend publication and hope that the recommendations of this paper are taken seriously
Author Response
Reviewer 3:
Comments and Suggestions for Authors
This is well-structured papers that makes a sound argument. As a feminist sociologist of religion, I did not find its main argument to be particularly novel and the modesty-concerns of religious women are common knowledge. However this paper does indeed make a valuable contributing in quantifying these concerns and making recommendations that are based on statistical information rather that qualitative data. I recommend publication and hope that the recommendations of this paper are taken seriously
Authors response:
We thank the reviewer for the critical reading, for the time and effort. Thank you for the excellent comments, and we will be happy to make further comments if necessary.